# Advice to Clinicians on Communication from Adolescents and Young Adults with Cancer and Parents of Children with Cancer

**DOI:** 10.3390/children10010007

**Published:** 2022-12-21

**Authors:** Meghana Srinivas, Erica C. Kaye, Lindsay J. Blazin, Justin N. Baker, Jennifer W. Mack, James M. DuBois, Bryan A. Sisk

**Affiliations:** 1Department of Pediatrics, Division of Hematology/Oncology, Washington University in St. Louis, 600 South Taylor Avenue, Suite 155, St. Louis, MO 63110, USA; 2Division of Quality of Life and Palliative Care, St. Jude Children’s Research Hospital, 262 Danny Thomas Place, Memphis, TN 38105, USA; 3Riley Hospital for Children at Indiana University Health, 705 Riley Hospital Dr, Indianapolis, IN 46202, USA; 4Dana-Farber/Boston Children’s Center for Cancer and Blood Disorders, Department of Pediatric Oncology, Harvard Medical School, 44 Binney Street, Boston, MA 02115, USA; 5Department of Medicine, General Medical Sciences, Washington University School of Medicine, Taylor Avenue Building 00154D, St. Louis, MO 63110, USA

**Keywords:** adolescent and young adult, pediatric oncology, communication, patient-centered care, cancer survivorship

## Abstract

Effective communication is integral to patient and family-centered care in pediatric and adolescent and young adult (AYA) oncology and improving healthcare delivery and outcomes. There is limited knowledge about whether AYAs and parents have similar communication preferences and needs. By eliciting and comparing communication advice from AYAs and parents, we can identify salient guidance for how clinicians can better communicate. We performed secondary analysis of semi-structured interviews from 2 qualitative communication studies. In one study, 80 parents of children with cancer during treatment, survivorship, or bereavement were interviewed. In the second study, AYAs with cancer during treatment or survivorship were interviewed. We asked AYAs and parents to provide communication advice for oncology clinicians. Using thematic analysis, we identified categories of advice related to three overarching themes: interpersonal relationships, informational preferences, and delivery of treatment, resources, and medical care. AYAs and parents provided similar advice about the need for compassion, strong connections, hopefulness, commitment, and transparent honesty However, AYAs placed additional emphasis on clinicians maintaining a calm demeanor.

## 1. Introduction

The diagnosis of cancer is both life-altering and life-threatening. Delivering adequate information about diagnosis, treatment and prognosis may increase trust among parents of children with cancer [1]. Effective communication is integral to patient and family-centered care in adolescent and young adult (AYA) oncology and improving healthcare delivery and outcomes [2,3]. Parents of children with cancer reported increased peace of mind, self-sufficiency, decreased guilt, and greater trust in the physician when physicians provided high-quality communication [4,5,6,7,8,9]. Transparent disclosure of outcomes, irrespective of prognosis increased parental hope [10]. 

Adolescence and young adulthood is a unique, transformative phase of development in moral, social, physical, cognitive, and emotional domains [11]. Their thought processes evolve from concrete to abstract and AYAs strive for independence from their parents to develop their own identities. When AYAs are diagnosed with cancer, it is disruptive to their development, education, and employment [12,13,14,15]. In one study of 523 AYAs who were recently diagnosed with cancer, over half of the participants reported unfulfilled information needs [16]. Prior studies show that AYAs with cancer prefer to be engaged in decision-making, receive transparent prognostic disclosure, and learn about long-term side effects [2,16,17,18]. As such, providing high-quality communication is important to support, engage, and empower AYAs to participate in their own care [2,16,18,19].

However, we know little about the communication needs and priorities of young AYAs with cancer, or how these priorities compare to parental perspectives. By seeking advice from AYAs and parents on communication, we aim to develop guidance for how clinicians can better communicate. Furthermore, comparing advice from AYAs and parents allows us to identify unique or differing priorities for clinician communication behaviors. In this secondary analysis of two qualitative datasets, we analyze advice from parents and AYAs in pediatric oncology about how clinicians should communicate with their patients and families. 

## 2. Materials and Methods

### 2.1. Participants and Recruitment

Parents of children with cancer were recruited from Washington University School of Medicine [St. Louis, MO, USA], Dana-Farber Cancer Institute [Boston, MA, USA], and St Jude Children’s Research Hospital [Memphis, TN, USA]. They were interviewed between October 2018 and March 2020. We used stratified sampling, aiming for 12–15 parents per stratum: time point [treatment ≥ 1 month, survivorship ≥ 6 months, or bereavement ≥ 6 months], child’s age at diagnosis [≤12 years or ≥13 years], and study site [20]. Parents were eligible if they were the parent most involved in communication with clinicians, had a child with cancer ≤18 years at the time of enrollment or death], and spoke English. We excluded participants who had clinical relationships with authors. We identified participants from review of patient lists, inpatient census, and outpatient schedules and recruited via telephone, postal mail, and in person. Institutional review boards at all sites approved this study.

AYAs with cancer from Washington University School of Medicine [St. Louis, MO, USA] and St. Jude Children’s Research Hospital [Memphis, TN] were recruited and interviewed between July 2020 and May 2021. We stratified sampling based on time point: patients undergoing treatment for at least 1 month [i.e., treatment] versus patients who completed treatment at least 6 months prior [i.e., survivorship]. We aimed for at least 15 participants per time point to ensure thematic saturation [21]. Eligibility criteria included cancer diagnosis, aged between 12 and 24 years at diagnosis, spoke English, and treated at pediatric facilities. The participants did not have clinical relationships with the authors. We recruited via telephone and acquired verbal consent. For minors, we obtained parental permission and participant assent. We explained the purpose of this communication study to the participants. Institutional review boards at all participating sites approved this study.

### 2.2. Data Collection

Data for this study was retrieved from two qualitative communication studies [2,22]. Semi-structured telephone interviews were conducted using interview guides. [Appendix A]. At the conclusion of each interview, we asked parents, “What about advice for the medical team after the child has finished their cancer treatments”. Similarly, AYAs were asked, “Imagine that doctors and nurses are going to tell a family that a child has cancer. What should these doctors and nurses be sure to do?”. All interviews were conducted by 1 of 2 authors [B.A.S. and L.J.B.]. Both were clinical fellows with qualitative research training. Interviews were audio-recorded and professionally transcribed.

### 2.3. Data Analysis

We performed thematic analysis on interview transcripts, assessing communication advice from AYAs and parents to oncology clinicians. Two authors [BAS and MS] developed the codebook by reading all transcripts to familiarize themselves to the content, then reviewing 5 to 10 transcripts in iterative cycles and independently developing descriptive codes and memos. The authors met to review codes, develop, and refine code definitions, and collapse redundant codes. We reached thematic saturation for advice to clinicians on communication after consensus coding of 37 AYA and 50 parent transcripts.

## 3. Results

### 3.1. Participant Characteristics

#### 3.1.1. Parent Characteristics

Eighty interviews with parents ranged from 24 to 108 min. Parents were predominantly White [91%]) and female [84%]). Diagnoses included leukemia or lymphoma [45%], solid tumors [39%], and brain tumors [16%]. (Table 1). Refer to appendix for graphical representation of participant characteristics.

#### 3.1.2. AYA Characteristics

Thirty-seven interviews with AYAs ranged from 21 to 89 min. AYAs were predominantly White [70%]. At the time of diagnosis, participants’ ages ranged from 12 to 20 years [mean = 16 years], and 28/37 participants were younger than 18 years at diagnosis. At the time of the interview, AYAs’ ages ranged from 13 to 25 years [mean = 18 years] (Table 2).

### 3.2. Overarching Themes of Communication Advice to Clinicians

We identified 3 overarching themes of communication advice for clinicians from AYAs and parents: maintaining interpersonal relationships, providing information, and providing treatment, resources, and medical care. The main themes and respective subthemes are listed in table in Table 3. Definitions of sub-themes are described in Table 4.

The major themes with corresponding sub-themes and representative excerpts from participants are described below.

#### 3.2.1. Maintaining Interpersonal Relationships

Advice to clinicians about maintaining interpersonal relationships focused on demonstrating empathy, maintaining optimism and hopefulness, empowering patients and families, building genuine connections, showing commitment, and maintaining a calm demeanor. 

##### Demonstrating Empathy

Most parents [68/80] emphasized the importance of physicians showing empathy, especially when the family is overwhelmed and distressed. Displays of empathy could be explicit or subtle. Some parents wanted clinicians to recognize the impact of their words, and transparent delivery of information needed to be balanced with empathy and compassion. “She was compassionate but straightforward. You can’t sugarcoat it…but have some warmth and compassion at the same time” [Mother, bereavement].

AYAs (17/37) similarly advised clinicians to show empathy by being kind, asking about their emotional state, and acknowledging the patient’s important role in the conversations. AYAs also advised clinicians to “read the room” before delivering bad news, ensuring that the patient and family were ready to receive this information. “It’s nice to have some form of showing that you actively care. I remember specifically the doctor sat down and he was close to my mom, and then my mom started to break down and then just put his hand on her shoulder and was, like, ‘Everything’s gonna be fine... We’re gonna get through this’” [Male, 18 y.o., survivorship]. Compassionate and kind clinicians relieved some AYA’s perceived emotional burdens during challenging times “There was just so much love somehow in that room at the same time. I just felt that she really loved me and wanted to see me just overcome it” [Female, 21 y.o., survivorship].

##### Showing Commitment

Parents [31/80] advised clinicians to demonstrate dedication and reliability in caring for their children. They advised clinicians to follow through on promises and to be there for them. For bereaved parents, this commitment should last after the child’s death. “Just continue to check on the family, even if it’s three months to six months after… I know it helped when I had his doctors and nurses show up at his viewing and funeral, to see that he wasn’t just a patient to him, that he did matter, to know that even though your child’s passed, you made an impact on these people’s lives” [Mother, bereavement]. 

AYAs [4/37] advised clinicians to make them feel like they were not alone and that the team remained available to them. “Be there and support them and do the best that they could do. Let the parent know that they’re there for them. That’s their job to be there and help in any way they can” [Male, 17 y.o, treatment]. AYAs in survivorship emphasized that clinicians should remain available to the patient after completing therapy, even if the AYA has transitioned to a different clinical team.

##### Empowering Patients and the Family

Parents [19/80] provided advice to clinicians on supporting and encouraging parents to advocate for their child and make patients/families feel like active members of the team. Parents advised empowering a child and his/her caregivers to ask questions. Parents of deceased children advised clinicians to support them in difficult decision-making, especially towards end-of-life. “I felt like I was being pushed to make decisions that I wasn’t comfortable with or ready to make.... I didn’t feel like they wanted to hear me” [Mother, bereavement]. A few parents advised that clinicians should talk to parents first before sharing information with the child. “Just keep in mind that they’re kids. The majority of information should go to the parents. The parents should be the ones who decide who tells the child and how much the child knows” [Mother, bereavement].

AYAs [10/37] advised clinicians to make them an important part of decisions and communication. AYAs appreciated when clinicians made them part of discussions. “Make sure that, like my doctor did, allow them to ask questions and stuff. Make sure that they feel informed about what’s going on because, obviously, it’s going on in their body. They would probably like to know what’s happening” [Female, 13y.o., treatment]. Some AYAs described that they still needed their parents and advised clinicians to appropriately incorporate their parents in communication. They described that their parents knew them best and had more experience in life.

##### Maintaining Optimism and Hopefulness

Parents [15/80] advised that clinicians should reinforce their hopes about their child’s future. These hopes could focus on prognosis or other specific hopes, highlighting the importance of hopefulness and positivity in general. “The last thing they wanna see is their mom or dad looking sad and gloomy. If it’s gonna be their last time on earth, let it be something happy, cheerful, and positive” [Father, bereavement]. Despite the importance of hopefulness, parents advised against providing false or unrealistic hopes. “As a medical professional, making a family aware of what’s going on and still instilling in them hope and personal care is important” [Mother, treatment].

AYAs [10/37] advised clinicians to be optimistic despite adversities. They advised clinicians to stay upbeat and positive instead of being gloomy. “I feel like you just have to have this really, really big presence of hope in the way that you speak” [Female, 19 y.o, survivorship]. While AYAs identified the importance of honesty and transparency, they also emphasized the importance of portraying information in a positive, hopeful manner. “The first thing would be to not look somber and not feel like—don’t give off a sad vibe or anything. I’d be positive... Be sympathetic. Be reassuring” [Male, 17 y.o, treatment].

##### Building and Maintaining Connections

Parents [9/80] advised clinicians to maintain a lasting relationship with families, even after their child has completed treatment. “Maybe the medical team could just offer their support. Hey, we’re here if you need us, but you are free to live normally again” [Mother, survivorship]. Bereaved parents advised clinicians to maintain a close-knit bond with them by providing avenues for continued communication after they lose their child. “They’re like family to them, cuz they’ve known ‘em so long. They’ve been through the worst times in their life. That is important to the family that he’s not forgotten, and that those people were really friendships with us, not just patient-doctor” [Mother, bereavement]. 

AYAs [10/37] advised clinicians to get to know them as a person and not just as a patient, often describing clinicians as “friends” or “family”. They advised clinicians to build a relationship with them by asking about their interests, being friendly, and displaying affection rather than being “sterile” and distant. “I think that they should ask more questions about what we liked doing or what are our habits or how we are and, so they get to know us a little bit more, even if they don’t necessarily remember because they have a lot of patients. It’s so nice to know that they listened to you”. [Female, 14 y.o., survivorship]. AYAs in survivorship advised clinicians to maintain a close-knit bond with them by providing avenues for continued communication after the end of treatment. 

##### Maintaining a Calm Demeanor

AYAs (12/37) advised clinicians to maintain a sense of calm, rather than showing their nervousness or anxiety. When their clinicians exhibited nervousness, it made them and their parents more anxious. They advised clinicians to make patients and families feel like things were under control. “You can be a little more laid back and just try to put off an aura of everything’s all good, but not too much ’cause you don’t wanna seem like that you’re not concerned about the kid’s health. You also don’t wanna put off an aura of too much anxiety ’cause then the parents are gonna pick up on that. The way they first came in, it was like, ‘This is pretty upsetting news’. I wasn’t that upset about it. You don’t have to talk like it’s very depressing” [Female, 15 y.o., treatment]. 

No parent provided advice on maintaining a calm demeanor.

#### 3.2.2. Providing Information

##### Supporting Understanding

Parents [59/80] advised clinicians on supporting understanding of information. Many parents advised that clinicians should try to understand each parent’s emotional state and learning preferences and adapt information sharing accordingly. “I think giving both written and verbal information is good because, I think, people are learners in different ways” [Mother, survivorship]. Parents advised that clinicians use simple terms, pace the amount and timing of information, and remain open to a multitude of questions without showing frustration. “The advice I have is to treat them like this is the very first time you’re having to explain it to a parent. Make eye contact. Check to make sure that they understand. To say, ‘Is there anything else I need to cover again for you?’” [ Mother, survivorship]. 

AYAs [19/37] advised clinicians to support understanding by providing comprehensible information about the diagnosis, treatment, prognosis, and of cancer and its treatment. AYAs advised that clinicians first recognize a family’s emotional state before sharing potentially distressing information. “Don’t throw a whole bunch of information at them at one time because obviously, it’s very stressful and just scary time. They’re not gonna be able to hold a whole bunch of information” [Female, 19y.o., treatment]. They advised clinicians to use simple terms, pace the amount of information, and not overwhelm patients and parents. Some AYAs preferred information to be shared upfront. “I’d be completely honest. Let people know what exactly it is ’cause it is their life. They deserve to know exactly to the full extent what’s going on with them” [ Male, 22 y.o., survivorship]. In contrast some of them preferred that distressing information be shared more gently and gradually. 

##### Preparing for the Future

Parents [20/80] advised clinicians to communicate when uncertainties present, dissipate them when there are answers r, and help patients and families to cope when unexpected situations arise. “Explain what the process is gonna look like and what the decisions that they’re gonna have to make will be, and what the path would be in any of those decisions” [Mother, bereavement]. Parents also advised clinicians to provide pragmatic information on post-treatment surveillance, treatment-related complications, relapse, and mortality to help parents set realistic expectations. “Long-term effects. What are some things you might see down the road? Just that constant monitoring because of the anxiety that you still carry with you when you have a cold or if you have a pain somewhere that you don’t think you’re supposed to have the pain, it makes you jump to the thought, ‘Is it back?’” [Mother, survivorship]. 

Only one [1/37] AYA provided advice on preparing for the future. “Acknowledging their fears of the future, at least for me, as a person who went through cancer, even though they told me I was cancer free, and I was in remission. There’s always this thought in mind, ‘Is it gonna come back? Is it gonna show up in a different cancer form? Acknowledging those feelings, maybe even bringing them up. I want the doctor to know that although it’s the end, in a cancer survivor’s mind, it’s not the end… that’s a fear with them all the time” [Female, 21 y.o., survivorship].

#### 3.2.3. Providing Treatment, Resources, and Medical Care

##### Providing Resources

Parents [15/80] provided advice on offering individualized resources and advised clinicians to offer them early on and continue to do so throughout their treatment and thereafter. These resources included Child Life Services, social workers, art therapy, music therapy, financial, and insurance resources. “I would’ve liked to have more follow-up with the social worker… Offer something besides a child life specialist, like maybe a psychologist. Anything just to tell us that she’s doing a great job. There really wasn’t much as far as mental health” [Mother, survivorship].

AYAs [3/37] advised clinicians to provide individualized resources to support self-management and emotional needs, early on. “I think providing resources for support is really important, and something that I didn’t get until later than I would have like” [Female, 18 y.o., treatment]. They emphasized that the clinicians should provide resources to connect with other AYAs who shared similar experiences. “I think that it’s really important to understand you’re not the only one. There are other people that are just being thrown back into this normal life that are struggling because they feel like they’re in between these two worlds, of this world where things used to be normal and you’re tryin’ to make it normal again, and this world where your world—everything in your life revolves around just making it through the day and staying well, and staying alive. That’s something I’m even dealin’ with today” [Male, 22 y.o., survivorship].

##### Being Competent

Parents [5/80] provided advice on clinicians demonstrating competency. Parents advised clinicians to display the knowledge, skill, and humility to provide optimal treatment for their children. “Know exactly what you’re talking about. There should be no questions in your mind when you’re having a conversation with a patient and families when it comes to a diagnosis and treatment” [Mother, survivorship].

One AYA [1/37] advised their clinicians to demonstrate the knowledge and skills to treat their cancer and its side effects. Further, they advised clinicians to fulfill their responsibilities towards their patients to the best of their abilities. “[Clinicians should say] we will do everything within our power and even possibly beyond our power to help your child get back to the health that they were at” [Male, 20.y.o, survivorship].

## 4. Discussion

AYAs with cancer and parents offered similar communication advice that aligned with three prominent themes related to clinician-patient relationships, information provision, and delivery of cancer care. Most advice related to clinical relationships, with an emphasis by parents and AYAs on showing kindness, commitment, and optimism. These findings reinforce prior studies demonstrating the centrality of relationships to effective communication in oncology [2,22,23]. However, these findings also highlight the challenges clinicians face in fulfilling these roles for patients, especially when tensions exist. For example, participants advised that clinicians should empower patients in their care, but also appreciate the important role of parents. Furthermore, clinicians should be honest and transparent, but maintain an optimistic demeanor. For many clinical encounters, these pieces of advice will not be in tension. However, clinicians will need additional guidance and training when these communication goals conflict. Consider the parents who do not want clinicians to disclose bad news to the adolescent, or AYAs who desire optimism and positive communication despite a poor prognosis [24]. Health care providers hesitate to refuse parental request for non-disclosure of diagnosis or prognosis to children, out of apprehension of destabilizing the family unit [25]. When confronting these types of conflicting goals and obligations, clinicians might consider engaging with trusted multidisciplinary colleagues, palliative care teams, or ethics consultants, depending on the scenario.

AYAs uniquely emphasized the importance of clinicians maintaining a calm demeanor and having a positive attitude without displaying nervousness. Past research has shown that parental anxiety is associated with an increased risk of developing anxiety and mood disorders in children. Nunn et.al, in their article ‘Keeping our children safe and calm in troubled times’, described how conveying the known and the unknown in a calm manner can instill confidence in children that will get them through challenging times [26]. Similarly, AYAs in this study seemed to rely on clinicians to keep them safe and demonstrate control and confidence by remaining calm. Providing reassurance that things are under control when faced with a life-threatening disease can be difficult. Yet, AYAs emphasized the importance of tone, nonverbal communication, and attitude, rather than the content alone. This finding is similar to past studies showing that many AYAs seek clear and transparent disclosure about diagnosis, treatment, and prognosis, yet feeling harmed by doctors who were patronizing, aloof, or cold [27,28].

Advice on ‘Preparing for the future was identified by 25% of parents, but only 1 (3/%) AYA. However, past studies have shown that AYAs value information about long-term toxicities and late effects [27,29]. It is possible that AYAs find this information important, but do not prioritize this information as highly as other aspects of communication. Developmentally, younger AYA patients often live in the present moment without fully considering future consequences. However, AYAs must be prepared for self-management of future complications from their cancer and its treatment, including physical, social, and emotional challenges that can present long after cancer treatment ends [30,31,32]. Clinicians might collaborate with parents and leverage existing transition programs to help AYAs become engaged in care and prepare for the future, such as the Got Transition six core elements of successful transition from pediatric to adult-focused care [33,34].

Advice on offering resources early and often during treatment and survivorship was identified in 19% [15/80] of parents and only 8% [3/37] AYAs. Many of these resources related to emotional and psychological support, in addition to financial resources. Both parents and their children can experience lasting financial and emotional sequelae of their child’s cancer [35,36,37]. The burden of these challenges can seem even more challenging after completion of therapy, when families have fewer interactions with the clinical team. In a previous study, many parents reported the need for more support during and after their child’s cancer treatment, and fewer than half of parents reported using existing support services [38]. Providing emotional support and enabling family self-management are two core functions of communication in pediatric and AYA oncology. Clinicians can support families by collaborating with social workers, child life specialists, psychologists, and patient advocacy groups to identify and refer to appropriate supportive services.

Our study adds to the growing body of literature highlighting the significance of high-quality communication in pediatric oncology. It emphasizes many of the nuances and challenges of navigating patient- and family-centered communication, especially when patients have relapsed or refractory disease. Despite the Accreditation Council for Graduate Medical Education (ACGME) having established communication as a core competency for physicians in training, pediatric hematology oncology trainees have limited communication training. Educational curricula should incorporate novel teaching methods and address communication challenges faced by trainees and educators [39,40].

This study has some limitations. AYAs were predominantly White or Black/African American. Hispanic adolescents were especially underrepresented, and interviews were only offered in English. Caregivers in our study had higher than average education level with almost 46% with college degrees [41]. Several studies have shown that higher parental education level is one of the factors associated with early cancer diagnoses and lower mortality rates in childhood cancer [42,43,44,45]. Cultures differences influence information sharing regarding a child’s illness and communication responsibilities [46]. Future research is suggested with a more diverse population, exploring experiences of adult survivors of childhood cancer are needed. Lastly, participants may have been affected by recall bias.

## 5. Conclusions

AYAs with cancer and parents of children with cancer provided advice to clinicians that focused on interpersonal relationships, information preferences, and delivery of treatment, resources, and medical care. AYAs and parents provided similar advice about the need for compassion, strong connections, hopefulness, commitment, and transparent honesty. AYAs also emphasized the need for clinicians to maintain a calm affect to ease their anxiety, and parents emphasized the need for clinicians to provide supportive resources to help them manage their child’s and family’s needs. Clinicians can follow this advice to better support patients and parents in pediatric and AYA oncology.

## Figures and Tables

**Table 1 children-10-00007-t001:** Patient and Parent Characteristics.

Parent Age	n (%)
21–29 years	4 (5)
30-39 years	25 (31)
40–49 years	31 (39)
50 years or older	20 (25)
**Parent gender**	
Female	67 (84)
Male	13 (16)
**Relation to Child**	
Parent	79 (99)
Grandparent	1 (1)
**Parent race/ethnicity ***	
White	73 (91)
Black	7 (9)
Asian	2 (3)
Hispanic	3 (4)
Other	1 (1)
**Parent education**	
High school graduate or less	7 (9)
Some college or technical school	15 (19)
College or technical school graduate	37 (46)
Graduate/professional school	21 (26)
**Parent marital status**	
Married/living as married	63 (79)
Other	17 (21)
**Child age at diagnosis**	
12 years or younger	52 (65)
13 years or older	28 (35)
**Child gender**	
Male	42 (53)
Female	38 (47)
**Diagnosis**	
Leukemia/Lymphoma	36 (45)
Solid tumor (not in brain)	31 (39)
Brain tumor	13 (16)
**Time point in cancer trajectory**	
Treatment	30 (37)
Survivorship	27 (35)
Bereavement	21 (27)
**Site**	
St Louis	27 (35)
Boston	27 (35)
Memphis	24 (30)

* Not mutually exclusive. Race and ethnicity were self-reported.

**Table 2 children-10-00007-t002:** AYA characteristics.

Gender	n (%)
Female	19 (51)
Male	18 (49)
**Age at diagnosis, mean (SD)**	16 (2.2)
**Age at interview, mean (SD)**	18 (2.9)
13—16 years	11 (30)
17—20 years	20 (54)
21—24 years	5 (14)
**Race ****	
Asian	2 (5)
Black	8 (22)
Pacific Islander	2 (5)
White	26 (70)
Hispanic ethnicity	1 (3)
**Diagnosis**	
CNS tumor	8 (22)
Leukemia	7 (19)
Lymphoma	12 (32)
Solid tumor	10 (27)
**Time point in cancer trajectory**	
Treatment	19 (51)
Survivorship	18 (49)
**Site**	
St Louis	19 (51)
Memphis	18 (49)

** Race and ethnicity were identified by chart review. Racial categories were not mutually exclusive, thus percentages total more than 100%.

**Table 3 children-10-00007-t003:** Three major themes and sub-themes.

	Parent Sub-Themes	AYA Sub-Themes
Interpersonal relationships	Demonstrate empathy and caring Showing commitment Empowering patients and families Maintaining hopefulness and optimism Building and maintaining connections	Demonstrate empathy and caring Showing commitment Empowering patients and families Maintaining hopefulness and optimism Building and maintaining connections Maintaining a calm demeanor
Informational preferences	Supporting understanding Preparing for the future	Supporting understanding Preparing for the future
Delivery of treatment, resources, and medical care	Providing resources Being competent	Providing resources Being competent

**Table 4 children-10-00007-t004:** Definitions of Subthemes.

Interpersonal Relationships		
Demonstrating empathy and caring	Parents experience profound emotional distress when their child is diagnosed with cancer. Emotional distress can impede their ability to process information and make informed treatment decisions. Parents advised that clinicians could help them through these challenging times by being compassionate, recognizing emotional needs, attempting to understand their feelings, and being responsive to each family’s unique needs. and celebrate achievements and important time points.	AYAs advised clinicians to ask about patients’ and families’ emotional states, acknowledge them, and respond appropriately. They advised clinicians to “read the room” before delivering a cancer diagnosis to meet the family’s unique emotional needs. Compassionate and kind clinicians can relieve some of their burdens during challenging times.
Showing commitment	Parents advised clinicians to demonstrate dedication and reliability in caring for their children. They advised clinicians to follow through on promises	AYAs advised clinicians make them feel like they were not alone and that the team remained available to them. AYAs in survivorship especially emphasized that clinicians remain available to the patient after completing therapy, even if the AYA has transitioned to a different clinical team.
Empowering the patient and family	Parents advised that clinicians should actively support and encourage parents to advocate for their child and make patients/families feel like active members of the team.	AYAs advised clinicians to make them feel like an important part of decisions and communication. They also advised clinicians to encourage AYAs to ask questions and make them comfortable to raise concerns.
Maintaining hopefulness and optimism	Parents advised that clinicians should reinforce their hopes about their child’s future. These hopes could focus on prognosis or other specific hopes. Many parents also described the importance of hopefulness and positivity in general. Despite the importance of hopefulness, parents also advised against providing false or unrealistic hopes.	AYAs advised clinicians to be optimistic despite adversities. They advised clinicians to stay upbeat and positive instead of being gloomy. While AYAs identified the importance of honesty and transparency, they also emphasized the importance of portraying information in a positive, hopeful manner.
Building and maintaining connections	Parents advised clinicians to maintain a lasting relationship with families, especially after their child has died. Similar to AYAs parents advised clinicians to maintain a close-knit bond with them by providing avenues for continued communication after they lost their child	AYAs advised clinicians to get to know them as a person and not just as a patient, often describing clinicians as “friends” or “family”. They advised clinicians to build a relationship with them by asking about their interests, being friendly, and displaying affection rather than being “sterile” and distant. AYAs in survivorship advised clinicians to maintain a close-knit bond with them by providing avenues for continued communication after the end of treatment.
Maintaining a calm demeanor	Not identified in parents	AYAs advised clinicians to maintain a sense of calm, rather than showing their nervousness or anxiety. When their clinicians exhibited nervousness, it made them and their parents more anxious. They advised clinicians to make patients and families feel like things were under control.
**Informational preferences**		
Supporting understanding	Parents pursue clarity regarding the cause, diagnosis, treatment, prognosis, and long-term effects of cancer and its treatment. Parents advised that clinicians use simple terms, pace the amount of information, and remain open to a multitude of questions without showing frustration. Further, many parents found that written information was helpful in addition to verbal communication.	AYAs also look for clarity in information regarding the cause, diagnosis, treatment, prognosis, and long-term effects of cancer and its treatment. AYAs advised that clinicians first recognize a family’s emotional state before sharing potentially distressing information. They advised clinicians to use simple terms, pace the amount of information, and not overwhelm patients and parents. Many AYAs preferred to know accurate percentages of pts with similar diagnoses and outcomes, but this was not universal.
Preparing for the future	Parents experience a variety of incertitude when the diagnosis of cancer is made. These uncertainties can stem from prognoses, frequency and or length of hospitalization, and late effects. Parents advised clinicians to communicate clinical uncertainty when it exists, dispel uncertainty when there is an answer, and help patients and families to cope when inexorable obscurities remain or arise. To help families navigate through these challenging situations, parents advised that clinicians should provide them with a plan for the immediate, short- and long-term futures. Further, providing pragmatic information on post-treatment surveillance, treatment-related complications, relapse, and mortality helped parents set realistic expectations.	AYAs advised clinicians to prepare for the future by actively acknowledging AYAs’ persistent worry about cancer relapse, and by preparing them to look for concerning signs or symptoms of illness or relapse. They also advised clinicians to help patients understand what to expect in survivorship, in terms of physical signs and symptoms, emotional distress, and logistics of follow-up with the clinical team.
**Delivery of treatment, resources, and medical care**		
Providing resources	Families advised clinicians to offer different resources to them early on and continue to do so throughout their treatment and thereafter. These resources included Child Life Services, social workers, art therapy, and music therapy, as well as financial and insurance resources. Parents shared that support staff helped them navigate through long hospital stays with support such as housing accommodations, gas cards, and meal card vouchers.	AYAs advised clinicians to provide individualized resources to support self-management and emotional needs. They emphasized that clinicians should provide resources to connect with other AYAs who shared similar experiences.
Being competent	Parents expect their clinicians to have the knowledge and skill to treat their cancer and its side effects effectively. Parents advised that clinicians demonstrate the knowledge, skill, and humility to provide optimal treatment for their child	AYAs similarly expected their clinicians to demonstrate the knowledge and skills to treat their cancer and side effects. AYAs advised clinicians to fulfill their responsibilities towards their patients to the best of their abilities.

## Data Availability

Not applicable.

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
