# Peer review of "Advice to Clinicians on Communication from Adolescents and Young Adults with Cancer and Parents of Children with Cancer"

_children, 2022, doi:10.3390/children10010007_

Round 1

Reviewer 1 Report

Overall, this is a well-written and helpful manuscript/study.

1. The authors report that the education level of parents in table 1 is higher than average (almost 46% with college degrees), however this is not highlighted in lines 368-378 when the "limitations" are discussed.

2. I would also suggest that neither the point of education nor the points discussed in lines 368-378 are true study limitations, they are the parameters in which this study was done which may (or may not) represent limitations of applicability. I only highlight this seemingly small point because when applying the authors findings to discussions with patients and their parents, it is probably more important to base the application on culture and education level than race, parental marital status, or a language barrier.

Author Response

Dear reviewer,

Thank you for your feedback and for bringing this to our attention.

  1. We agree that the proportion of parents with college degrees is higher than the average. This point is included in the discussion below.
  2. We agree that cultural differences and the education level of parents are factors that influenced the experiences of participants in our study.

This study has some limitations. AYAs were predominantly White or Black/African American. Hispanic adolescents were especially underrepresented, and interviews were only offered in English. Caregivers in our study had higher than average education levels with almost 46% with college degrees. Cultural differences influence information sharing regarding a child’s illness and communication responsibilities. Future research is suggested with a more diverse population, exploring experiences of adult survivors of childhood cancer are needed.  Lastly, participants may have been affected by recall bias.

Reviewer 2 Report

In the manuscript “Advice to clinicians on communication from adolescents and 2 young adults with cancer and parents of children with cancer” Srinivas et al., presented effective communication is integral to patient and family-centered care in pediatric and adolescent and young adult (AYA) oncology and improving healthcare delivery and outcomes. Though the sample size is small and from a single center, the overall study was impressive.

Minor Comments:

Overall study was straightforward, however, even though the data was shown in the table, pie chart, or Venn diagram for the Patient and Parent Characteristics and the output, would be impressive. 

Author Response

Dear reviewer,

Thank you for your feedback. We have incorporated bar charts for some of the main participant variables. Please find attached document for graphical representation.
